# Direct and indirect punishment of norm violations in daily life

Catherine Molho [1,2 ✉], Joshua M. Tybur[1], Paul A. M. Van Lange[1] & Daniel Balliet[1]

Across societies, humans punish norm violations. To date, research on the antecedents and consequences of punishment has largely relied upon agent-based modeling and laboratory experiments. Here, we report a longitudinal study documenting punishment responses to norm violations in daily life ($k = 1507$; $N = 257$) and test pre-registered hypotheses about the antecedents of direct punishment (i.e., confrontation) and indirect punishment (i.e., gossip and social exclusion). We find that people use confrontation versus gossip in a context-sensitive manner. Confrontation is more likely when punishers have been personally victimized, have more power, and value offenders more. Gossip is more likely when norm violations are severe and when punishers have less power, value offenders less, and experience disgust. Findings reveal a complex punishment psychology that weighs the benefits of adjusting others' behavior against the risks of retaliation.

[1] VU Amsterdam, Department of Experimental and Applied Psychology, Institute for Brain and Behavior Amsterdam (IBBA), Van der Boechorststraat 7, 1081BT Amsterdam, the Netherlands. [2] Institute for Advanced Study in Toulouse, Esplanade de l'Université 1, Toulouse 31080, Cedex 06, France. ✉email: catherine.molho@iast.fr

A key challenge to the evolution and maintenance of cooperation involves deterring cheaters and regulating norm violations. Communities and individuals regularly face norm-violating behaviors (e.g., free-riding and littering[1]). In response, people are often motivated to impose costs on offenders via punishment[2–4]. Such punishment can confer benefits to those who mete it out (e.g., deterrence, status, and reputation)[5–8] and to their groups (e.g., resource preservation and public goods provision)[9–11]. Despite its ubiquity and consequences, little is known about the factors underlying punishment in natural settings.

Theoretical accounts suggest that cooperation is maintained via direct reciprocity, indirect reciprocity, and/or partner choice[12]. These accounts argue that various forms of punishment—costly punishment[2,9,11], gossip[13,14], and social exclusion[15,16]—can effectively deter cheating and promote norm abidance[9,17,18]. However, most empirical observations on punishment come from interactions in laboratory settings[9,11,19] which, while well-controlled, lack many aspects of the real-world ecologies in which punishment occurs. In more ecologically valid settings, a host of factors—relational, situational, and emotional—can influence the use of punishment.

When do people deploy different forms of punishment? One possibility is that, when norm violations are detected, people use distinct punishment strategies in an unconditional manner. That is, people may randomly choose among punishment strategies or merely use those strategies that are available. Another possibility is that, upon detecting a norm violation, people enact specific punishment strategies, conditional on factors that shift the costs and benefits of punishment. To illustrate, direct punishment strategies, which are overt and involve physical or verbal confrontation of offenders, are risky[20,21], because they expose one to retaliation[22]. At the same time, direct punishment can swiftly remove threats and effectively adjust offenders' behavior[5,23]—whether by physically deterring offenders or by verbally communicating disapproval and condemnation to them[24].

In contrast, indirect punishment strategies such as gossip and social avoidance[13–15] are less risky. Gossip, exclusion, and avoidance can be employed in the absence of the offender and hence involve lower costs than confrontational punishment, in that they are less likely to elicit retaliation[25]. Such indirect behaviors are often intended to impose costs on offenders[21,25,26], and they involve a host of negative experiences for their targets[27,28]. Besides their phenomenological costs, social exclusion and avoidance restrict offenders' access to benefits offered by coalitional allies (including the punisher). That said, indirect strategies may be less effective than confrontational ones at promptly stopping violations (e.g., if gossip spread is slow or if offenders are merely avoided).

If people indeed use direct versus indirect punishment strategies in a conditional manner, what sort of decision rules do they employ? First, decision rules underlying punishment should be fine-tuned to the benefits of adjusting others' behavior, which may be higher when (a) the offender is a highly-valued individual[29] or (b) the violation is victimizing oneself (rather than someone else)[30–32]. Second, decision rules underlying punishment should be tailored to minimizing the risks of retaliation from offenders, which may be higher when (c) violations are more severe[33] or (d) offenders are more powerful[34]. Of course, such cost-benefit calculations need not be conscious; (e) negative emotions may be a primary motivator of punishment[9,35,36].

To test these hypotheses, we document punishment of norm violations—i.e., behaviors that participants considered immoral, unacceptable, or improper[37,38] (for more details, see "Methods" section)—via daily assessments over the course of 2 weeks[39]. While prior experimental work has typically studied costly punishment (i.e., economic sanctioning) in interactions between strangers, we instead capture a broader range of high- and low-cost punishment responses to norm violations occurring within various relationships. To do so, we measure direct (i.e., physical and verbal confrontation) and indirect (i.e., gossip and exclusion/avoidance) responses to norm violations, which can be used to impose material and/or reputational costs on offenders—be it through physical aggression, verbal communication, reputation manipulation, and/or the withdrawal of social benefits[21,24,34,40]. Further, we assess both motivations to punish (i.e., what people felt like doing) and punishment behaviors (i.e., what people actually did)[41,42]. Using two follow-up surveys, we report longer-term patterns by assessing punishment responses one to 2 weeks after the violations occurred.

Based on reports of norm violations ($k = 1507$) and follow-up responses ($k = 311$) from a community sample of 257 Dutch participants (66% female, age range: 18–75 years), we test pre-registered hypotheses regarding the relational (i.e., valuation of offenders and victim of violations), situational (i.e., moral wrongness and power), and emotional (i.e., anger and disgust) antecedents of punishment. We find that people punish norm violations in daily life, employing confrontation, gossip, and social avoidance in context-sensitive ways. Confrontation is more likely when punishers have more to gain—i.e., when they value offenders more and when they have been personally victimized by norm violations. In contrast, gossip and social avoidance are more likely when the costs of potential retaliation loom large—i.e., when violations are severe and when offenders possess more relative power. Anger is associated with harsher punishment across the board, whereas disgust is specifically associated with more indirect punishment. Together, our findings show that people consider both the benefits of changing others' behavior and the costs of potential counter-punishment, when deciding how to punish in daily life.

## Results

**Frequency of punishment strategies in daily life**. Existing theories propose that costly punishment[2,9,11], gossip[13,14], and social exclusion[15] represent key strategies for promoting cooperation. The few field studies on punishment further suggest that low-cost strategies, such as withholding benefits, may be more frequent than costly, confrontational ones[21]. So far, observations in the field mostly speak to punishment in interactions between strangers[21,33,41]. What are the strategies people use to address violations within various relationships—family, friendship, work—in their daily life?

Figure 1a shows the distributions of motivations to engage in different types of punishment in daily life. Participants differentially endorsed motivations to engage in various types of punishment ($k = 1236$; $F(3, 2537.27) = 111.19$, $p < 0.001$). Specifically, they expressed stronger motivations to punish via gossip ($M = 2.82$, $SD = 1.24$) and social exclusion ($M = 2.79$, $SD = 1.37$) rather than via physical ($M = 2.13$, $SD = 1.18$) or verbal confrontation ($M = 2.45$, $SD = 1.28$) (see Supplementary Table 1).

Further, Fig. 1b, c show the frequencies of punishment behaviors in daily and in follow-up assessments. In daily assessments, participants differentially engaged in various types of punishment behaviors (Wald $\chi^2(2) = 27.64$, $p < 0.001$). Specifically, they were more likely to gossip (in 44.1% of events) than to directly confront (in 35.4% of events; Wald $\chi^2(1) = 11.37$, $OR = 1.33$, $p = 0.001$) or avoid offenders (in 34.8% of events; Wald $\chi^2(1) = 24.65$, $OR = 1.45$, $p < 0.001$). However, when participants were physically present, and were thus able to directly confront offenders, there was no difference (Wald $\chi^2(1) = 0.20$, $OR = 1.05$, $p = 0.658$) in the prevalence of gossip

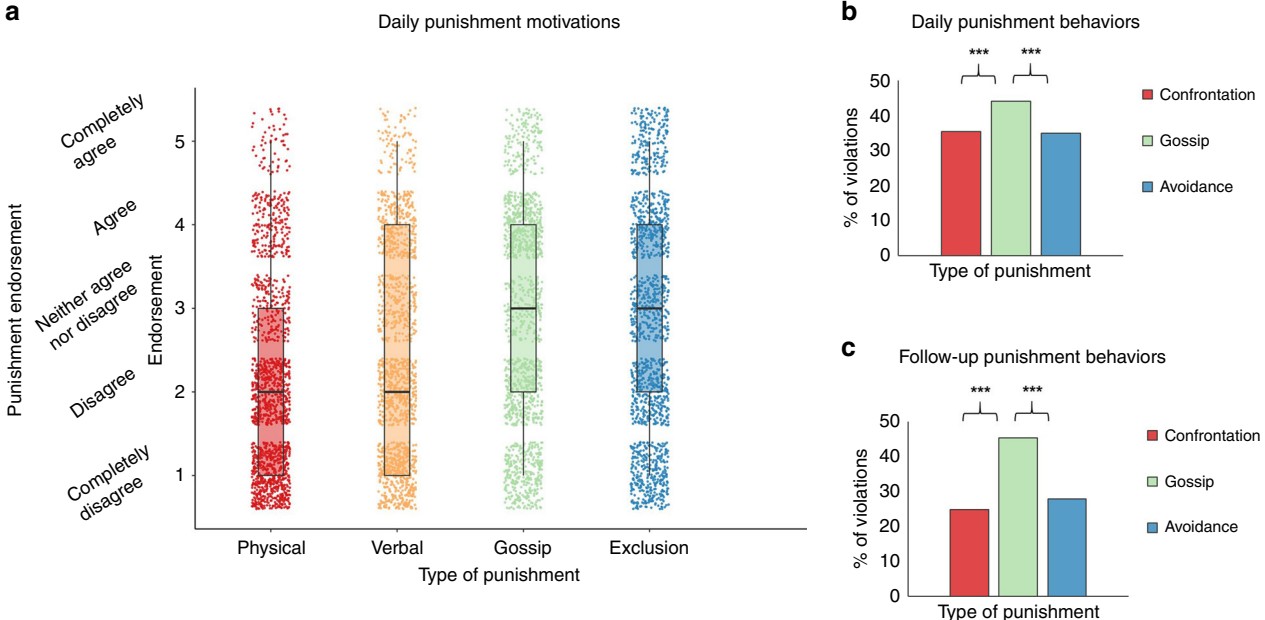

**Fig. 1 Prevalence of different types of punishment responses in daily life. a** Participants' endorsement of motivations to engage in various types of punishment, based on $k = 1236$ daily assessments. Four items assessed motivations to punish offenders via physical confrontation ("I felt like physically intervening to stop the offender."), verbal confrontation ("I felt like yelling at or arguing with the offender."), gossip ("I felt like sharing negative information about the offender to others."), and exclusion ("I felt like excluding the offender from my social interactions in the future."). Boxplot whiskers indicate the minimum (1) and maximum (5) values observed, box bounds indicate the first quartile (equal to 1 for physical and verbal confrontation; equal to 2 for gossip and exclusion) and third quartile (equal to 3 for physical confrontation; equal to 4 for other types of punishment motivations), and horizontal lines indicate the median. Error bars represent standard deviations from the mean. **b** Percentages of violations in $k = 1236$ daily assessments and **c** in $k = 879$ follow-up assessments to which participants responded with each type of punishment behavior. Bars represent the percentage of assessments where participants responded "Yes" to items measuring confrontation ("I confronted the offender about his/her behavior."; daily: 35.4%, follow-up: 24.9%), gossip ("I told someone else about this behavior when the offender was absent."; daily: 44.1%, follow-up: 45.4%), and avoidance ("I avoided social contact with the offender."; daily: 34.8%, follow-up: 27.9%). Results are based on Generalized Estimating Equations models with punishment type (confrontation, gossip, and social avoidance) as a factor and punishment behavior as the outcome (daily: Wald $\chi^2(2) = 27.64$, $p < 0.001$; follow-up: Wald $\chi^2(2) = 145.81$, $p < 0.001$). Planned contrasts were performed, without adjustments for multiple comparisons. All tests were two-sided. ***indicates $p$ values $\leq 0.001$. Source data are provided as a Source data file.

(45.9% of events) and confrontation (42.6% of events). Instead, both gossip and confrontation were more likely than avoidance (37.4% of events; gossip vs. avoid: Wald $\chi^2(1) = 12.07$, $OR = 1.37$, $p = 0.001$; confront vs. avoid: Wald $\chi^2(1) = 5.61$, $OR = 1.31$, $p = 0.018$).

Relative to elicitation methods that ask participants to recall events over a longer time frame[39], our use of daily assessments reduces recall bias. However, this method might underestimate the frequency of punishment, if sufficient time had not elapsed for people to confront or gossip about offenders. To address this issue, we examined punishment behaviors in follow-up assessments that were completed 7–14 days after violations took place. Results revealed the same pattern of punishment behaviors as in daily assessments (see Supplementary Methods for details). Finally, in the Supplementary Methods, we report patterns of punishment behaviors in situations that more closely align with laboratory approaches to assessing second-party punishment (i.e., self-relevant offenses perpetrated by strangers) and third-party punishment situations (i.e., offenses perpetrated by strangers and targeting strangers). We note here that direct confrontation is substantially rarer in situations that resemble third-party punishment tasks, consistent with previous work[20,43].

**Punishment is sensitive to benefits of changing others' behavior.** One important function of punishment is to adjust offenders' behavior in a way that promotes the interests of the punisher[5,23,44]. The benefits accrued by punishers may, in turn, depend on the closeness[42] and valuation (i.e., welfare tradeoff ratio)[29,43,44] of their relationship with offenders. Importantly, then, the use of direct and indirect punishment strategies may vary based on the valuation of offenders. Individuals may directly confront offenders whom they value highly, because (a) adjusting highly-valued others' behavior can accrue greater benefits than adjusting less-valued others' behavior and (b) there is less uncertainty regarding close others' response to punishment. Additionally, people may hesitate to gossip about or avoid valuable interaction partners, because such behaviors can be particularly detrimental within long-term relationships[35,45].

The relation between participants' valuation of offenders and their endorsement of punishment motivations and behaviors varied across different types of punishment (motivations: $F(3, 4253.25) = 19.92$, $p < 0.001$; behaviors: Wald $\chi^2(2) = 74.42$, $p < 0.001$). Specifically, participants were less motivated to punish offenders whom they valued more ($F(1, 697.57) = 15.69$, $p < 0.001$), and this negative relation was stronger for motivations to exclude offenders compared with motivations to engage in any other type of punishment (all $ps \leq 0.001$, see Table 1 and Supplementary Table 2). Further, when participants valued offenders more, they were more likely to engage in direct confrontation (Wald $\chi^2(1) = 38.44$, $b = 0.16$, $p < 0.001$), whereas they were less likely to gossip (Wald $\chi^2(1) = 13.40$, $b = -0.09$, $p < 0.001$) or engage in social avoidance (Wald $\chi^2(1) = 28.94$, $b = -0.14$, $p < 0.001$). Together, findings support the idea that direct confrontation is more frequently used against highly-valued offenders, whereas gossip and social exclusion are more frequently deployed against less-valued offenders.

**Table 1 Valuation of offenders and endorsement of punishment motivations.**

|  | Estimate | t | df | p |
|---|---|---|---|---|
| Intercept | 3.11 | 57.51 | 3473.32 | <0.001 |
| Physical | −0.96 | −15.33 | 4704.45 | <0.001 |
| Verbal | −0.53 | −8.60 | 4594.26 | <0.001 |
| Gossip | 0.03 | 0.48 | 4288.59 | 0.632 |
| $WTR_{own}$ (person-centered) | −0.16 | −6.51 | 1072.91 | <0.001 |
| Physical × $WTR_{own}$ (person-centered) | 0.10 | 6.09 | 4641.87 | <0.001 |
| Verbal × $WTR_{own}$ (person-centered) | 0.12 | 7.08 | 4436.94 | <0.001 |
| Gossip × $WTR_{own}$ (person-centered) | 0.05 | 3.35 | 4029.86 | 0.001 |

Results from a linear mixed model with punishment type, $F(3, 4427.47) = 112.76$, $p < 0.001$, participants' valuation of offenders ($WTR_{own}$), $F(1, 697.57) = 15.69$, $p < 0.001$, and the $WTR_{own}$ × punishment type interaction, $F(3, 4253.25) = 19.92$, $p < 0.001$, as predictors of punishment motivations. The table shows parameter estimates from planned contrasts, without adjustments for multiple comparisons. All tests are two-sided. Social exclusion motivations are used as the reference category. The model controls for gender and the gender × punishment type interaction. The model also includes the effects of $WTR_{own}$ (person-average), and the $WTR_{own}$ (person-average) × punishment type interaction. Results including person-average effects are available in Supplementary Table 2. Source data are provided as a Source data file.

**Table 2 Moral wrongness of norm violations and endorsement of punishment motivations.**

|  | Estimate | t | df | p |
|---|---|---|---|---|
| Intercept | 1.26 | 6.38 | 3519.08 | <0.001 |
| Physical | 0.14 | 0.59 | 5838.62 | 0.552 |
| Verbal | 0.02 | 0.09 | 5551.19 | 0.930 |
| Gossip | 0.45 | 2.02 | 3532.99 | 0.043 |
| Moral wrongness (person-centered) | 0.51 | 10.88 | 3217.97 | <0.001 |
| Physical × moral wrongness (person-centered) | −0.13 | −2.71 | 4261.51 | 0.007 |
| Verbal × moral wrongness (person-centered) | −0.14 | −3.06 | 4386.70 | 0.002 |
| Gossip × moral wrongness (person-centered) | −0.12 | −2.58 | 3796.70 | 0.010 |

Results from a linear mixed model with punishment type, $F(3, 4215.43) = 1.78$, $p = 0.148$, moral wrongness, $F(1, 1471.93) = 120.06$, $p < 0.001$, and the moral wrongness × punishment type interaction, $F(3, 4008.72) = 3.92$, $p = 0.008$, as predictors of punishment motivations. The table shows parameter estimates from planned contrasts, without adjustments for multiple comparisons. All tests are two-sided. Social exclusion motivations are used as the reference category. The model controls for gender and the gender × punishment type interaction. The model also includes the effects of moral wrongness (person-average), and the moral wrongness (person-average) × punishment type interaction. Results including person-average effects are available in Supplementary Table 3. Source data are provided as a Source data file.

Findings thus far are consistent with the view that people upregulate their punishment when there is more to gain from adjusting offenders' behavior. However, punishment might also function to enforce moral norms, even in the absence of personal benefit[2,11]. Drawing upon an altruistic punishment perspective[9,11], some research suggests that people punish norm violations that victimize others more harshly than they punish violations victimizing themselves[46]. However, evidence from more ecologically valid settings so far points to the opposite pattern[37,42]. In addition, multiple recent vignette studies[31] find that victims of norm violations (i.e., second-parties) are more motivated to directly punish offenders than are third-party observers, whereas second- and third-parties are similarly motivated to indirectly punish offenders.

Here, motivations to engage in various types of punishment were similar, regardless of whether participants were victimized by or merely observed the norm violations ($F(3, 2640, 35) = 1.66$, $p = 0.174$). However, consistent with predictions, participants engaged in different types of punishment behaviors depending on the self-relevance of violations (Wald $\chi^2(2) = 19.97$, $p < 0.001$). Being personally victimized by violations was associated with substantially more direct confrontation (Wald $\chi^2(1) = 84.98$, $b = 1.07$, $p < 0.001$). Indeed, when participants were the victims of violations, compared to being observers, they were more likely to confront rather than gossip about (Wald $\chi^2(1) = 16.74$, $OR = 1.86$, $p < 0.001$) or avoid offenders (Wald $\chi^2(1) = 15.90$, $OR = 1.92$, $p < 0.001$). In sum, motivations to engage in different types of punishment seem unaffected by the self-relevance of violations, but people are much more likely to follow through with punishment—especially via directly confrontation—when they are personally victimized.

**Punishment is sensitive to the risks of retaliation.** Existing accounts suggest that punishment may be deployed proportionately to the severity of norm violations[9,11]. Under this view, people will more harshly punish violations that are seen as more morally wrong[42] and as deviating more from the contribution standards in their group[11]. However, the severity of norm violations may also be used as a cue to the risks of retaliation from offenders. Offenders that have previously engaged in severe violations may be perceived as more committed and able to counter-punish. If so, direct punishment may be avoided when violations are judged as more severe or morally wrong[33] and, instead, people may use more indirect strategies to minimize retaliation from perpetrators of severe violations.

Here, the relation between moral wrongness and participants' endorsement of punishment motivations and behaviors varied across different types of punishment (motivations: $F(3, 4008.72) = 3.92$, $p = 0.008$; behaviors: Wald $\chi^2(2) = 20.62$, $p < 0.001$). Moral wrongness was more strongly, positively associated with motivations to socially exclude offenders compared with motivations to engage in any other type of punishment (all $ps \leq 0.01$; see Table 2 and Supplementary Table 3). Further, when responding to violations judged as more morally wrong, participants were more likely to socially avoid (Wald $\chi^2(1) = 12.06$, $b = 0.25$, $p = 0.001$) or gossip about offenders (Wald $\chi^2(1) = 10.22$, $b = 0.23$, $p = 0.001$), but less likely to confront them (Wald $\chi^2(1) = 4.48$, $b = -0.15$, $p = 0.034$). Together, results support the idea that severe violations are met with more gossip and social exclusion, whereas direct confrontation appears less likely in the face of violations perceived as more morally wrong.

These findings are consistent with the proposition that punishment strategies are conditioned upon the risk of retaliation

from offenders. Examining how individuals' power relative to offenders influences distinct punishment strategies offers a more direct test of the idea that punishment varies as a function of offenders' ability to impose retaliation costs. Being in a relatively high-power position—in terms of having access to valued resources or the ability to impose costs[44,47]—may lower people's threshold to directly punish offenders. Conversely, being in a low-power position may increase sensitivity to the risk of retaliation from offenders[47] and incentivize lower-cost strategies (gossip, ridicule, exclusion)[34] to deter violations.

The relation between participants' power relative to offenders and their endorsement of punishment motivations and behaviors varied across different types of punishment (motivations: $F(3, 4247.63) = 2.75$, $p = 0.041$; behaviors: Wald $\chi^2(2) = 42.19$, $p < 0.001$). When participants had lower power, they reported stronger motivations to exclude offenders ($b = -0.16$, $p = 0.002$). The (negative) relations of power with physical confrontation and gossip motivations were of similar strength, but the association between power and motivations to verbally confront offenders was much weaker ($p = 0.011$; see Table 3 and Supplementary Table 4). Further, when participants had lower power, they were more likely to engage in gossip (Wald $\chi^2(1) = 5.28$, $b = -0.17$, $p = 0.022$) or avoidance (Wald $\chi^2(1) = 4.23$, $b = -0.16$, $p = 0.040$), but less likely to engage in confrontation (Wald $\chi^2(1) = 36.21$, $b = 0.48$, $p < 0.001$). In sum, findings support the idea that low-power individuals use gossip and exclusion to deter cheating or exploitation by the powerful, whereas high-power individuals use more confrontational strategies of punishment.

**Punishment strategies are motivated by distinct emotions**. Existing work suggests that the punishment of norm violations is often motivated by intuitive, affective reactions to offenders. Yet, this work has often treated various moral emotions—anger, disgust, and contempt—as equivalent[9,36,42]. Nevertheless, recent attempts to disentangle moral emotions based on their functional consequences have proven fruitful[30,31,35], showing evidence of unique associations between anger and motivations to engage in direct punishment and between disgust and motivations to engage in indirect punishment. Indeed, while anger has traditionally been viewed as motivating approach and confrontation[30,44], theoretical accounts have seen disgust as motivating social distancing and efforts to coordinate punishment of offenders[48].

Feelings of anger did not differentially relate to distinct punishment motivations ($F(3, 4277.69) = 2.17$, $p = 0.089$) or behaviors (Wald $\chi^2(2) = 3.38$, $p = 0.185$). Instead, anger was associated with stronger punishment across the board

(motivations: $F(1, 372.55) = 67.16$, $p < 0.001$, behaviors: Wald $\chi^2(1) = 7.62$, $p = 0.006$). Feelings of disgust were differentially associated with motivations to engage in various types of punishment ($F(3, 4166.67) = 3.79$, $p = 0.010$) and with distinct punishment behaviors (Wald $\chi^2(2) = 11.43$, $p = 0.003$). When participants felt more disgust, they reported stronger motivations to gossip about and exclude offenders, compared with motivations to physically confront them (for both contrasts, $p$s < 0.01; see Table 4 and Supplementary Table 5). Disgust did not differentially relate with motivations to physically and verbally confront offenders ($p = 0.445$). Similarly, when participants felt more disgust, they were more likely to engage in gossip (Wald $\chi^2(1) = 11.43$, $OR = 1.32$, $p = 0.001$) and avoidance (Wald $\chi^2(2) = 3.98$, $OR = 1.20$, $p = 0.046$), compared to direct confrontation. This pattern of results remained when controlling for general emotional state (ranging from very negative to very positive, cf. ref. [49]). Overall, we found that anger positively relates to both direct and indirect punishment, but we observed stronger relations of disgust with indirect strategies of gossip and social exclusion, rather than direct confrontation.

## Discussion

This study set out to document people's direct and indirect punishment of norm violations in natural settings. Consistent with evolutionary models[10,12], as well as experimental studies[9,11,13,15], we observed widespread use of various strategies to punish offenders, including direct confrontation, gossip, and social avoidance. Overall, gossip was the most frequent response to violations, although confrontation was just as likely when immediate intervention was possible. Both direct confrontation and gossip were more prevalent than avoidance; this may reflect the differential costs and benefits of these strategies. While confrontation is characterized by its immediacy and effectiveness in changing offenders' behavior, gossip has the potential to manipulate offenders' reputation without affecting the relationship between the gossiper and the target (assuming the identity of the gossiper remains concealed from the target). Thus, avoidance may be least prevalent because it is (a) less beneficial than confrontation in terms of adjusting others' behavior, and at the same time (b) more costly than gossip in terms of lost interaction opportunities or damage to one's relationship with offenders.

While it is possible that, upon detecting norm violations, people use multiple punishment strategies in an unconditional manner, our findings instead suggest that direct and indirect punishment are highly context-dependent. We found that people were more likely to punish violations that personally victimized them (compared to someone else), and especially so by directly confronting offenders. Further, people did not unconditionally

**Table 3 Relative power and endorsement of punishment motivations.**

|  | Estimate | t | df | p |
|---|---|---|---|---|
| Intercept | 3.25 | 26.13 | 3733.10 | <0.001 |
| Physical | −0.95 | −6.50 | 5828.31 | <0.001 |
| Verbal | −0.54 | −3.72 | 5162.47 | <0.001 |
| Gossip | 0.12 | 0.89 | 3411.62 | 0.376 |
| Power (person-centered) | −0.16 | −3.12 | 3594.75 | 0.002 |
| Physical × power (person-centered) | 0.06 | 1.20 | 4637.48 | 0.228 |
| Verbal × power (person-centered) | 0.13 | 2.53 | 4717.07 | 0.011 |
| Gossip × power (person-centered) | 0.01 | 0.14 | 3936.69 | 0.885 |

Results from a linear mixed model with punishment type, $F(3, 4066.10) = 22.51$, $p < 0.001$, power, $F(1, 1635.80) = 7.18$, $p = 0.007$, and the power × punishment type interaction, $F(3, 4247.63) = 2.75$, $p = 0.041$, as predictors of punishment motivations. The table shows parameter estimates from planned contrasts, without adjustments for multiple comparisons. All tests are two-sided. Social exclusion motivations are used as the reference category. The model controls for gender and the gender × punishment type interaction. The model also includes the effects of power (person-average), and the power (person-average) × punishment type interaction. Results including person-average effects are available in Supplementary Table 4. Source data are provided as a Source data file.

**Table 4 Anger, disgust, and endorsement of punishment motivations.**

|  | Estimate | t | df | p |
|---|---|---|---|---|
| Intercept | 1.22 | 6.86 | 2814.86 | <0.001 |
| Verbal | −0.78 | −3.62 | 4292.11 | <0.001 |
| Gossip | 0.04 | 0.22 | 3679.08 | 0.824 |
| Social exclusion | −0.35 | −1.70 | 4145.23 | 0.088 |
| Anger (person-centered) | 0.19 | 4.79 | 970.36 | <0.001 |
| Verbal × anger (person-centered) | 0.09 | 1.97 | 4756.23 | 0.049 |
| Gossip × anger (person-centered) | 0.05 | 1.05 | 4605.54 | 0.293 |
| Social exclusion × anger (person-centered) | 0.10 | 2.31 | 3992.02 | 0.021 |
| Disgust (person-centered) | 0.12 | 3.26 | 1175.49 | 0.001 |
| Verbal × disgust (person-centered) | 0.03 | 0.70 | 4633.02 | 0.483 |
| Gossip × disgust (person-centered) | 0.11 | 2.75 | 4533.95 | 0.006 |
| Social exclusion × disgust (person-centered) | 0.10 | 2.61 | 3873.57 | 0.009 |

Results from a linear mixed model with punishment type, $F(3, 3970.60) = 6.69$, $p < 0.001$, anger, $F(1, 372.55) = 67.16$, $p < 0.001$, disgust, $F(1, 450.90) = 42.18$, $p < 0.001$, and the anger × punishment type, $F(3, 4277.69) = 2.17$, $p = 0.089$, and disgust × punishment type, $F(3, 4166.67) = 3.79$, $p = 0.010$, interactions as predictors of punishment motivations. The table shows parameter estimates from planned contrasts, without adjustments for multiple comparisons. All tests are two-sided. Physical confrontation motivations are used as the reference category. The model controls for gender and the gender × punishment type interaction. The model also includes effects of anger (person-average) and disgust (person-average), as well as the anger (person-average) × punishment type and disgust (person-average) × punishment type interactions. Results including person-average effects are available in Supplementary Table 5. Source data are provided as a Source data file.

increase all forms of punishment in proportion to the severity of offenses. Instead, they selectively up-regulated gossip and social avoidance, rather than direct confrontation, when responding to severe violations. Together, these findings provide support for the idea that punishment is deployed in ways that promote punishers' interests[5,23,44].

Further, findings support the idea that one important function of punishment is to deter future transgressions[23,44]. We found that people were more likely to directly punish highly-valued offenders—whose behavioral adjustments can provide more future benefits to the punisher—and more morally wrong offenses—which may indicate potential recidivism. That said, punishment can promote the interests of punishers in multiple other ways, including via enhancing their status and reputation[6–8,50]. Evidence that individuals were more likely to directly punish self-relevant violations is indeed consistent with the idea that people punish to defend their reputation, even when deterrence concerns are irrelevant (e.g., in one-shot games)[11,40,50]. Evidence that high-power individuals were more likely to directly confront offenders, whereas low-power individuals used more indirect punishment, also points to the fact that punishment may serve competitive goals, aiming to increase one's advantage over subordinates[40,44].

Finally, findings described here provide support for socio-functional accounts of moral emotions[30,31,35,51,52], according to which anger and disgust toward offenders are associated with unique functional consequences (e.g., aggressive motivations). In this study, when people experienced more anger, they responded with more punishment across the board. In contrast, when they experienced more disgust toward offenders, they consistently deployed more indirect, rather than direct, punishment. This latter finding supports recent theory proposing that disgust motivates social distancing from offenders and efforts to recruit punishment from others[31,48].

This study extends our understanding of punishment in natural settings by examining a broad range of high- and low-cost strategies to punish norm violations within various types of relationships. One caveat of the more inclusive definition of punishment employed here—which encompasses physical threats, verbal condemnation, reputation manipulation, and social exclusion—is that it errs on the side of considering behaviors that have not effectively inflicted costs on offenders. We believe that this limitation is countered by the benefits of the rich, naturalistic information obtained via our methodology. That said, future work would benefit from directly measuring the costs

imposed on offenders via distinct punishment strategies. Moreover, the prospective data collection method we have employed here has the benefit of reducing recall bias[39]. At the same time, it could have made participants more attentive to norm violations occurring in daily settings, while also increasing the overall frequency of their interventions. Importantly, this issue does not affect our conclusions regarding how relational, situational, and emotional factors relate to the use of direct versus indirect strategies of punishment in daily life.

On the whole, this study provides evidence that people deploy direct and indirect punishment in a conditional manner, depending on (a) the benefits of changing offenders' behavior, (b) the risks of receiving retaliation, and (c) their underlying emotional states. This evidence has implications for theoretical models on the evolution of cooperation and on the enforcement and spread of norms. Specifically, findings suggest that modeling work would benefit from considering variation in distinct punishment strategies and in contextual factors that relate to punishment, including power dynamics, social network properties, and the costliness of others' offenses. More generally, findings highlight that not all punishments are created equal. Experimental studies have often subsumed different strategies under the umbrella of 'costly punishment'. Yet, in daily life, the punishment of norm violations takes a multitude of forms. Considering the antecedents of distinct punishment strategies can help explain long-observed phenomena, such as limited bystander intervention in the face of severe offenses. It can also help address challenges that only recently appeared in our social world, such as the widespread expression of moral outrage and indirect punishment via social media.

## Methods

**Materials, data, and code.** Methods and materials for the study were pre-registered and are available on the Open Science Framework (OSF; registration DOI [https://doi.org/10.17605/OSF.IO/FDZXT]). The data and syntax that are relevant to the analyses described herein are also publicly available[53], DOI [https://doi.org/10.17605/OSF.IO/DU7MP].

**Sample and data collection.** We used two Dutch panel agencies (Flycatcher and Link2Trials) to recruit 257 participants for a study with three parts: (a) an intake that took place in the laboratory of the VU Amsterdam; (b) a daily assessment phase, during which participants received daily surveys on their mobile phones for a 2-week period; and (c) a follow-up phase, in which participants were asked additional questions about events reported in the daily assessments, 7–14 days after they occurred. Among the 256 participants who provided demographic information, 66.1% identified as female (one participant chose the "other" option). The sample was diverse in terms of age ($M = 39.15$ years, $SD = 16.02$, range: 18–75

years) and was highly educated (60.5% had a bachelor's or higher degree; 39.5% had secondary education, including vocational training). The average subjective socioeconomic status (measured via a 1 to 10-point ladder method) was above the scale midpoint ($M = 6.49$, $SD = 1.54$). Most participants (90.2%) were born in the Netherlands.

This research was approved by the Research Ethics Committee of the VU Amsterdam (#VCWE-2018–052). All participants provided informed consent.

**Procedure**. Participants signed up via an online survey, which provided information about the study and the inclusion criteria (age ≥18, fluency in Dutch, owning a smartphone with internet access). They provided their contact details and indicated three time slots when they would be available to participate in the study. Then, they were contacted via email to schedule the intake.

*Intake*. Intake sessions took place in the laboratory of the VU Amsterdam in June and July 2018; sessions were conducted by the first author. Upon arrival, participants were led to closed cubicles and were asked to read and sign informed consent forms. In a fixed sequence, they completed: (a) questionnaires measuring individual differences (in randomized order), (b) demographic questions, (c) measures of physical formidability, and (d) incentivized, decision-making tasks (in randomized order). All questions and tasks were implemented in Qualtrics. Finally, participants received standardized, video-recorded instructions for the daily assessment phase (the transcript is available on the OSF; [https://doi.org/10.17605/OSF.IO/FDZXT]) and had the opportunity to ask clarification questions. On average, the intake took one hour to complete.

*Daily assessment phase*. Daily surveys were implemented in Qualtrics and sent to participants' mobile phones via SurveySignal[54]. The daily assessment phase started on the day after the intake and lasted for 2 weeks. On each day, at 19:00, participants received a text message with a link to the daily survey. If they did not complete the survey within an hour, they received a reminder. Each survey remained open for six hours. The median time until opening the link was one hour and one minute, and the median time of survey completion was 7 min.

In each daily survey, participants were first asked (a) whether they were personally affected by a behavior that they thought was wrong (Branch 1: self-relevant event) and then (b) whether they witnessed or learned about someone else being affected by a behavior that they thought was wrong (Branch 2: other-relevant event). Thus, in each daily survey, participants could report on a maximum of two violations: one victimizing themselves and one victimizing someone else. If participants responded "Yes" to both questions, they completed the respective questionnaires and were then sent to the end of the survey. If they responded "No" to one of these questions, they were then asked to (c) think about and report on the last situation that they experienced with another person (Branch 3: social event). Finally, if they responded "No" to both initial prompts, they were also asked to (d) think about and report on the last situation that they experienced alone (Branch 4: non-social event). Thus, participants reported on one out of four possible combinations of events: (1) self-relevant event & other-relevant event; (2) self-relevant event & social event; (3) other-relevant event & social event; (4) social event & non-social event.

The overall response rate was 80.27%. In total, participants completed 2888 daily surveys. In 1236 cases (42.80%), participants reported experiencing at least one violation. Given that it was possible to report on two violations in the same daily survey, the total number of reported violations was higher: $k = 1,468$ (self-relevant: $k = 901$; other-relevant: $k = 567$).

*Follow-up phase*. Follow-up surveys were also implemented in Qualtrics and sent to participants' phones via text messages. In the follow-ups, participants read the descriptions of norm violations that they had provided in the daily assessment phase, and they answered additional questions about their consecutive responses to those violations. We sent the first follow-up one day after the end of the daily assessment phase; there, participants answered questions about events from the first week of the daily assessment phase. We sent the second follow-up survey 1 week after the end of the daily assessment phase; there, participants answered questions about events from the second week of the daily assessment phase. Thus, there was a time-lag of between 7 and 14 days between daily surveys and follow-up surveys. As with the daily assessments, we sent participants a reminder if they did not respond to the follow-ups within an hour. Each follow-up survey remained open for 24 h.

The response rate for the first follow-up was 71.98%; the response rate for the second follow-up (40.02%) was lower. Due to a technical problem with the software used to automatically send surveys, 55 participants did not receive the second follow-up. When considering only participants who received the survey, the response rate for the second follow-up (62.38%) was higher. The median time until opening follow-up survey links was one hour and 20 min, and the median time of survey completion was six minutes.

*Participant compensation*. Participants received €20 for the intake, which included €5 for transportation costs. During the incentivized decision-making tasks in the intake, they could earn a bonus of up to €5. For each completed survey in the daily assessment phase, they received €1. Additionally, they could earn a bonus of €10 for completing at least 80% of the daily assessments. Finally, completion of each follow-up survey was incentivized with €5. In total, participants could therefore earn 20 (intake) + 5 (intake bonus) + 14 (daily assessments) + 10 (daily assessments bonus) + 10 (follow-ups) = €59. On average, participants received €49.35.

**Measures**. A list of all measures and materials is available on the OSF pre-registration page for the study: https://osf.io/fdzxt. In what follows, we describe in detail only measures and materials that are relevant to the present investigation.

*Intake*. At the end of the intake, participants received detailed, video-recorded instructions for the daily assessment phase (the full transcript is available on the OSF).

Specifically, we told participants that, in each daily survey, we would ask them about two types of events, both of which concern another person's behavior that they thought was wrong. We specifically instructed them to think about behaviors that they witnessed on the day of the assessment and that they thought were immoral, unacceptable, or improper[37,38]. We described these behaviors as going against their values or principles[55], and as potentially leading to disapproval and punishment from themselves or others[11].

We told participants that, in each daily survey, we would first ask them about events in which another person's wrong behavior personally affected them, and we provided examples of such behaviors (e.g., a friend lying, an acquaintance saying something offensive). We then told them that we would separately ask about events in which they witnessed or learned about another person's wrong behavior that affected someone else. Again, we provided various examples of such behaviors (e.g., witnessing a fight on the street, hearing something bad about their boss). Participants were instructed to think of situations in which they were physically present, but also about behaviors that they learnt about from another source. We mentioned that the offender could be someone they knew or a stranger, and that the violation could be serious or mundane. If on a given day they encountered more than one violation, we asked them to report on the one they found most important.

Participants were then shown screenshot examples of questions from the different parts of the survey (e.g., concerning their relationship with the offender and the victim, their emotions). Given our focus on punishment responses, we gave more detailed instructions regarding items measuring motivations to engage in various types of punishment and punishment behaviors. Specifically, we told participants that we would ask them questions about how they felt like reacting to the offender's behavior and gave them examples of items measuring motivations to engage in punishment. Then, we told them that we would ask them how they actually reacted to the offender's behavior. Here, we used examples to illustrate direct versus indirect punishment and emphasized their distinction based on whether they are overt (i.e., happened in the presence of the offender) or covert (i.e., happened in the absence of the offender). Specifically, for direct confrontation, we instructed participants to think of any behavior they did in the presence of the offender and in response to the wrong behavior ("Did you do something to confront the offender, such as physically stopping this person or arguing with him/her? Here, you can think of any behavior you did in the presence of the offender and in response to the wrong behavior."). For gossip, we instructed them to think of any information they shared with others in the absence of the offender and in response to the wrong behavior ("Did you tell someone else about the behavior of the offender? Here, you can think of any information you shared with others in the absence of the offender and in response to the wrong behavior."). We informed them that we would also ask whether they avoided social contact with the offender.

Participants also received brief information about the other branches of the daily surveys (Branches 3 & 4: social and non-social events). Finally, they learnt about the details of the compensation scheme, including the bonus for completing more than 80% of daily surveys, and had the opportunity to ask clarification questions.

*Daily assessment phase*. When participants indicated that they experienced a norm violation—either self- or other-relevant—we asked them to give a brief description of what had happened and to include information about the offender in their description. Examples of described violations are provided in the Supplementary Methods. We also asked participants to indicate whether they were physically present ($k = 968$) or not ($k = 500$) when the violation occurred. For a research question that is unrelated to the current investigation, we further measured the number of bystanders present in norm violations.

We then asked additional questions regarding the offender. We measured the type of relationship participants had with the offender (family member, romantic partner, friend, classmate or co-worker, instructor or supervisor, acquaintance, stranger, or other) and the gender of the offender (man, woman, other, I don't know). We also measured the emotional closeness (i.e., "At this moment, I feel close to the offender") and valuation of participants' relationship with the offender. To assess the latter, we used a measure of welfare tradeoff ratio (WTR$_{own}$)[29,43] toward the offender. Specifically, we asked participants to indicate what would be the highest amount of money (€0–10) they would forego for the offender to receive €10. We used the same method to assess participants' perceptions of the offender's welfare tradeoff ratio (WTR$_{other}$) toward them.

Further, we used two questions to assess the moral wrongness (1 = not at all morally wrong, 5 = extremely morally wrong) and harmfulness (1 = not at all harmful, 5 = extremely harmful) of the offender's behavior. Including both questions allowed us to explore the overlap between judgments of moral wrongness and harm in response to violations[56,57]. We also measured perceived interdependence with the offender, using three items from the Situational Interdependence Scale[58,59]. Situational power was measured with one item ("Who had the most influence on what happened in that situation?"; 1 = definitely the offender, 5 = definitely myself).

Only when participants indicated that they experienced an other-relevant norm violation, we asked them questions about the victim. Specifically, we measured the type of relationship they had with the victim and we asked for the victim's gender. We also assessed the emotional closeness of participants' relationship with the victim (i.e., "At this moment, I feel close to the victim"), their WTR$_{own}$ toward the victim and their perceptions of the victim's WTR$_{other}$ toward them. Finally, we measured participants' perceived interdependence with the victim, using items from the Situational Interdependence Scale[58,59].

To measure emotions, we used arrays of facial emotional expressions from the Radboud Faces Database[31,60] and asked participants to indicate whether these faces matched their feelings toward the offender (1 = completely disagree, 5 = completely agree). In this way, we measured five emotions (anger, disgust, fear, sadness, and happiness). We also used one item to assess the general valence of participants' emotional experience (1 = very negative, 5 = very positive).

Finally, we measured participants' motivations to engage in various types of punishment and their punishment behaviors. To measure motivations to engage in punishment, we adapted four items from previous work[31] assessing tendencies to physically ("I felt like physically intervening to stop the offender") or verbally ("I felt like yelling at or arguing with the offender") confront the offender, and to negatively gossip about ("I felt like sharing negative information about the offender to others") or socially exclude ("I felt like excluding the offender from my social interactions in the future") the offender. These items were rated on 5-point Likert scales (1 = completely disagree, 5 = completely agree). To measure punishment behaviors, we asked participants to indicate their agreement (binary scale: Yes or No) with three statements about how they actually reacted to the offender's behavior. We measured (1) direct confrontation ("I confronted the offender about his/her behavior."), (2) gossip ("I told someone else about this behavior when the offender was absent."), and (3) social avoidance ("I avoided social contact with the offender."). Finally, we included an open-ended question where participants could describe in detail how they behaved in response to the violation.

*Follow-up phase.* In the follow-up phase, we presented participants with each of the descriptions of norm violations that they had reported in the 2 weeks of the daily assessment phase. We instructed them to read these descriptions (and some additional information they had provided about the offender, i.e., their relationship type and the offender's gender) and to answer additional questions about them. Specifically, we used the follow-up surveys to re-assess participants' feelings of emotional closeness with the offender, their WTR$_{own}$ toward the offender, and their perceptions of the offender's WTR$_{other}$ toward them. We also assessed emotional responses to violations with the same five arrays of emotional expressions that we used in the daily assessment phase, and the additional item measuring general emotional state.

Then, we presented participants with the descriptions they gave about their behavioral responses to each of the violations from the daily assessment phase. We instructed them to answer questions concerning what they did on the days after the violation. Then, we assessed punishment behaviors (i.e., direct confrontation, gossip, and social avoidance) with the same three items used in the daily assessment phase. We again included an open-ended question where participants could describe in detail how they behaved on the days after the violation.

**Statistical analyses**. Data manipulation and data analyses were performed in R and in SPSS.

*Data exclusion procedures.* We did not exclude any participants from the analyses, but we only analyzed data from completed daily assessments and follow-up surveys. In some cases, due to technical reasons, participants were able to complete the same daily assessment or follow-up survey more than once. When they did so, we retained the first response. This was either the most complete response or, in few cases, one of multiple incomplete responses.

*Data analyses procedures.* All reported statistical tests are two-sided. In all analyses, we used models that account for the hierarchical structure of our data (i.e., reports of violations nested within days, nested within subjects). When predicting punishment motivations (continuous DVs, rated on 5-point Likert scales), we used linear mixed models (run via MIXED in SPSS) with random intercepts and slopes for days and subjects. When predicting punishment behaviors (binary DVs), we used binary logistic regression models (run via Generalized Estimating Equations in SPSS), again nesting observations within days and subjects. In both types of models—MIXED and GEEs—we specified an autocorrelation matrix, to account for the fact that measures taken closer in time can be more correlated than measures taken further in time.

In models including continuous IVs, we tested for relationships between punishment and both within-person-centered variables and person-average variables. Further, in linear mixed models, we specified additional random intercepts and slopes for within-person-centered variables (when IVs were continuous) and for binary IVs (without any transformation). Finally, given well-established gender differences in direct aggression[61], we controlled for participant gender in all analyses.

**Reporting summary**. Further information on research design is available in the Nature Research Reporting Summary linked to this article.

## Data availability

All data that are relevant to the analyses described herein are available on the OSF[53] (https://osf.io/du7mp/). A reporting summary for this Article is available as a Supplementary Information file. Source data are provided with this paper.

## Code availability

Syntax to reproduce the analyses described herein is available on the OSF (https://osf.io/du7mp/).

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

## Acknowledgements

The authors thank Junhui Wu and members of the Amsterdam Cooperation Lab for valuable suggestions on the manuscript. We thank Tim Alkemade, Margriet Bentvelzen, Marloes Doeswijk, Terence Dores Cruz, Maaike Homan, and Sterre van Niekerken for translations of study materials. C.M. acknowledges IAST funding from the French National Research Agency (ANR) under grant ANR-17-EURE-0010 (Investissements d'Avenir program). C.M. and D.B. were supported by a European Research Council grant (ERC StG-2014-635356). J.M.T. was also supported by the ERC (ERC StG-2015-680002-HBIS).

## Author contributions

C.M., J.M.T., P.A.M.V.L., and D.B. all participated in designing the study and drafting the paper. C.M. collected and analyzed the data.

## Competing interests

The authors declare no competing interests.
