## [Peer Review File · Nature Communications]

Reviewers' comments:

Reviewer #1 (Remarks to the Author):

In this paper, the authors report the results of an ambitious longitudinal study to measure punishment (and punishment motivations) in everyday life and test whether different punishment strategies are deployed under different circumstances. I think this paper has the potential to have a large impact in the field given its uniqueness of the data and method, which overcomes a lot of the shortcomings typical to laboratory punishment experiments. So, I think it's very important work, the methods and analyses are sound, and the paper is very well written. That being said, I do think there are some things the authors could address that would sharpen up the manuscript. To be clear: I think these are all things that can be addressed in a revision and I would love to see this paper published.

The biggest thing that sticks out to me as needing improvement is how the authors define and refer to punishment both in describing it and how they have constructed their measures. On p.3, they refer to gossip and social avoidance as "indirect punishment strategies." While I agree with that classification for gossip, social avoidance seems to fall more under partner choice than punishment unless the offender relies on benefits from the "punisher" that are then removed in retaliation. Simply avoiding future exploitation from someone shouldn't be classified as punishment. On p. 5 the authors refer to costly punishment, gossip, and social exclusion as separate things, but then lump them all together as punishment motivations in Fig. 1 and in their discussions of them. So, I think these things need to be clarified a bit in the text to make sure they are referring to consistent concepts.

On a related note, and more importantly, the motivations/actions that participants respond to don't match up as well to how the cooperation literature defines punishment, and how the authors themselves (whom I agree with) define punishment on p. 4, "...impose material and/or reputational costs on offenders." For reference, here are the phrasings for each item participants responded to from the supplemental material:

Motivations: "I felt like physically intervening to stop the offender," "I felt like yelling at or arguing with the offender," "I felt like sharing negative information about the offender to others," "I felt like excluding the offender from my social interactions in the future."

Actions: "I confronted the offender about his/her behavior," "I told someone else about this behavior when the offender was absent," "I avoided social contact with the offender."

For one, physically intervening to stop the offender doesn't necessarily mean there is a punishment motive—if one simply breaks up a fight between two strangers we wouldn't consider that punishment. What material or reputational cost is being imposed?

More concerning to me, though, is the direct confrontation question for whether someone took action. Merely confronting an offender about their behavior, without any other qualifiers, doesn't constitute punishment. Looking at some of the examples of norm violations reported (more on this below) causes me a bit of concern. Several of them mention someone being late or dropping out of contact—I really don't see how merely confronting a person about that (e.g., telling them you don't appreciate wasting your time, they'd better not do that again in the future, etc.) constitutes punishment. What cost is being imposed? Certainly a confrontation *could* include punishment (verbally attacking the person, withdrawing benefits in retaliation, etc.), but without further clarification I'm pretty unconvinced that just "confrontation" maps onto the construct of punishment focused on in the cooperation literature and what, I presume, the authors here are really interested in.

One way around this problem could be to incorporate the free response questions that participants answered about what action they took, which would clarify when punishment actually occurred. Without doing something along those lines, I think it's a bit misleading to characterize all forms of confrontation as punishment—if not, perhaps the authors could instead refer to direct confrontation and mention that punishment is a subset of those actions. Going this latter route, the paper could be reframed a bit more toward, say, strategies to recalibrate social norm violations, rather than solely focusing on "punishment."

Other reactions/questions:

- From my reading of the methods and the supplemental material, it seems like the authors did not exclude any cases from their analyses on the basis of the type of norm violation reported. If this is true, I am a bit concerned about the relevancy of some of the examples of social norm violations that are reported in the supplemental material. Some examples that caught my eye: "I'm in a hotel and filled up my 2 bottles of water during the breakfast buffet. I heard a man at the table next to me say to his table companions that this wasn't allowed", "I was working when a man without shirt came to ask things. Some other guests were looking uncomfortable", "When I was parking this afternoon, some other driver deemed it necessary to honk loudly. He probably thought I wasn't fast enough", "He tried to flirt with me and I didn't take it kindly", and "My partner wanted to be of too much help." Perhaps there are some cultural differences at play here (I'm American), but I find examples like these to be a decent stretch from even what I would colloquially call a social norm violation, much less a *cooperative* social norm violation that much of the cooperation literature is focused on, which is how the authors set the context of the paper in the introduction.

- Why did the authors distinguish between physical and verbal motivations for punishment but lump these together for behavior?

- The authors raise a good point that their daily sampling method can reduce recall bias relative to recall methods. I totally agree. However, the prospective design that the authors use here does have a weakness that they don't address: by instructing participants at the outset that they will need to report on norm violations and their responses to them every day, I would imagine there is some effect almost like an intervention. Particularly given that taking action in response to norm violations is socially desirable, I would suspect that the design employed here both (a) caused participants to be more vigilant for norm violations in their everyday life than they would normally, and (b) be more likely to take action than they normally would be since they would know they would have to report a lack of action at the end of the day if they didn't. Given that the main focus of the paper is on the use of different punishment strategies under different circumstances (i.e., the meat of the paper is looking at differences among contexts, not absolute levels of punishment), I don't think this is a problem that undermines the main findings. I do think it's worth mentioning, however, that it's a possible limitation—and I do also think it has likely resulted in higher levels of action taken than one would observe more naturally (i.e., without participants being conscious of their intervention behavior being monitored).

- Very minor, but some of the language used to describe motivations is hard to parse. One particular example on p.6, "We found that the strength of motivations varied across types of punishment" makes it seem like they are referring to motivations either in response to types of punishment or across different scenarios. Really, they are talking about motivations to engage in different types of punishment, so I'd consider rephrasing (I kept running into this in the results sections in both the main text and supplemental). Again, not a big issue but I think it could make the manuscript clearer to touch it up a bit.

To reiterate, I think this is a very important contribution to the literature and should be quite high impact if the authors are given the opportunity for a revision.

Reviewer #2 (Remarks to the Author):

This is an excellent paper, with a unique dataset and important results on a hotly debated issue in the many communities that study the evolution of cooperation and its psychological foundations (especially: evolutionary biologists, anthropologists, and psychologists; behavioral economists; cultural evolution modelers). Whether people punish others, when, and how is a central issue in these debates, and the answer matters to theories of whether cooperation evolved via reciprocity (and by partner choice versus partner control, i.e., punishment), via "strong reciprocity"/ group selection, via cultural group selection, or via some other population level process. This paper provides data that richly speaks to these theories. It also contributes to theories of the function of different emotions. On top of all this, it is beautiful and clearly written. It should be published with high priority.

Whether people punish at all in real life (as opposed to in anonymous situations in a laboratory) has been questioned: observational studies are usually limited to interactions involving strangers; when the studies are of small scale societies (as in anthropology), one does not know whether people are failing to punish for prudential reasons (they do not want to be retaliated against) or because they lack the motivation to do so.

The authors of this study have overcome all of these limitations. They have an ecologically valid data set, based on what happened to a large sample of people in their daily life. They collected the data in a systematic manner that does not rely heavily on memory; they collected two weeks of data from each subject; they found out what offense the subject was reacting to; and the subject's relationship to the offender (Someone they value highly? Are close to? Have situation power over?). They distinguished between theoretically important types of punishment (direct confrontation: physical or verbal; indirect punishment through gossip or exclusion). Importantly, for each offense, they assessed the subject's motivation to punish (e.g., I felt like yelling at him), as well as their actual behavior (I yelled at him). By assessing emotional reactions, they were also able to test predictions about the adaptive function of different emotions (e.g., that anger is a system that motivates bargaining for better treatment; that disgust motivates social withdrawal).

The authors were so thorough that they were able to discover which social and emotional variables predict different types of punishment, separating out fear of retaliation from how important the person is in your life, how wrong you thought the offense was, whether the offense harmed you or someone else, and so on. This paper has many riches: (i) a valuable dataset that measures psychological, social, and behavioral factors in a large, ecologically valid sample, (ii) the clarity of the authors' thinking about alternative hypotheses (re, e.g., strong reciprocity, "altruistic" third party punishment, dyadic reciprocity), and (iii) elegant statistical analyses with pre-registered hypotheses. My own view is that their data is most consistent with theories emphasizing adaptations that evolved by individual selection for dyadic or small group reciprocity. Others may disagree, but one thing is sure: this paper is going to be highly cited, and be used by all the above communities to evaluate alternative theories.

My recommendation to publish is unconditional: the authors do not need to change anything. Below, I have a few thoughts (and minor typos).

Main paper:

Line 146 When you refer to the "second party", I assume you mean the victim of the offense (with the offender as the first party). If so, you might want to say that in parentheses.

Line 192 This is just my curiosity. On the weaker association between power and motivation to confront, I wonder what would happen controlling for closeness/ how valuable the offender is to the subject. Is the weaker association because you are more motivated to confront people you are close too? (another result of yours)

Lines 474, 476 run (not ran)

SI 202 "they endorsed less indirect compared to confrontational motivations" The use of less here makes a little garden path. Maybe: they endorsed indirect motivations less than confrontational ones.

Reviewer #1

That being said, I do think there are some things the authors could address that would sharpen up the manuscript. To be clear: I think these are all things that can be addressed in a revision and I would love to see this paper published.

We thank the reviewer for the positive evaluation of our work. We appreciate the suggestions, which helped us clarify the conceptualization and operationalizations of punishment in our paper.

The biggest thing that sticks out to me as needing improvement is how the authors define and refer to punishment both in describing it and how they have constructed their measures. On p.3, they refer to gossip and social avoidance as “indirect punishment strategies.” While I agree with that classification for gossip, social avoidance seems to fall more under partner choice than punishment unless the offender relies on benefits from the “punisher” that are then removed in retaliation. Simply avoiding future exploitation from someone shouldn’t be classified as punishment.

The distinction we draw between direct and indirect punishment is based on seminal work differentiating direct versus indirect aggression (Archer & Coyne, 2005; Campbell, 1999). Based on this work, we posit that the key distinction between direct and indirect strategies of punishment lies in the fact that the former are overt and confrontational—i.e., they occur in the offender’s presence and are addressed to him/her—whereas the latter are covert and less confrontational— i.e., they occur behind the offender’s back. This emphasis is reflected in the instructions we provided for the daily assessment phase, and the way we illustrated confrontation (“*Did you do something to confront the offender, such as physically stopping this person or arguing with him/her? Here, you can think of any behavior you did in the **presence** of the offender and in response to the wrong behavior.*”) and gossip (“*Did you tell someone else about the behavior of the offender? Here, you can think of any information you shared with others in the **absence** of the offender and in response to the wrong behavior.*”).

In various literatures, social exclusion and avoidance are conceptualized as indirectly aggressive responses (or *covert, relational, social* aggression; Archer & Coyne, 2005; see also Kerr et al., 2009; Williams, 2007), because they do not require the presence of the offender, and because they involve lower costs than confrontation. This does not negate the key role that exclusion and avoidance play in partner choice. To the reviewer’s main point, prior work suggests that exclusion and avoidance are often intended to impose costs on offenders (e.g., Balafoutas et al., 2014; Campbell, 1999) and they involve a host of negative experiences for their targets (e.g., Eisenberger et al., 2003; Williams, 2007). Besides its phenomenological costs, social exclusion can restrict offenders’ access to benefits offered by coalitional allies (e.g., social support, protection, reciprocal benefits; Kerr et al., 2009; Yamagishi et al., 1999). In a similar vein, avoidance can be

detrimental for offenders, if only by restricting access to the benefits otherwise offered by the individual now avoiding them.

We have made multiple edits to the manuscript to clarify (a) how we differentiate direct and indirect forms of punishment (pp. 3-4), (b) how we made this conceptual distinction clear to participants when instructing them (pp. 20-21), (c) why we subsume social exclusion and avoidance under the category of indirect punishment (p. 4), and (d) what we see as the costs of social exclusion and avoidance for offenders (p. 4).

On p. 5 the authors refer to costly punishment, gossip, and social exclusion as separate things, but then lump them all together as punishment motivations in Fig. 1 and in their discussions of them. So, I think these things need to be clarified a bit in the text to make sure they are referring to consistent concepts.

We thank the reviewer for pointing out that the broad term ‘punishment motivations’ impeded, rather than clarified, the presentation of our findings. We have revised the manuscript to communicate that we *separately* consider different motivations to use various types of punishment.

On a related note, and more importantly, the motivations/actions that participants respond to don't match up as well to how the cooperation literature defines punishment, and how the authors themselves (whom I agree with) define punishment on p. 4, "...impose material and/or reputational costs on offenders." For reference, here are the phrasings for each item participants responded to from the supplemental material:

Motivations: "I felt like physically intervening to stop the offender," "I felt like yelling at or arguing with the offender," "I felt like sharing negative information about the offender to others," "I felt like excluding the offender from my social interactions in the future."

Actions: "I confronted the offender about his/her behavior," "I told someone else about this behavior when the offender was absent," "I avoided social contact with the offender."

For one, physically intervening to stop the offender doesn't necessarily mean there is a punishment motive—if one simply breaks up a fight between two strangers we wouldn't consider that punishment. What material or reputational cost is being imposed?

We agree that our operationalization of punishment does not strictly align with experimental studies on cooperation (e.g., studies using second- and third-party punishment games). Such studies have typically focused on interactions between strangers, either examining responses to self-relevant offenses in which the perpetrators are strangers, or examining responses to other-relevant offenses in which both perpetrators *and* victims are strangers. Further, experimental studies have tended to focus on specific expressions of punishment (in particular, non-cooperation and monetary sanctioning) in a well-specified, but somewhat limited, set of situations. Clearly,

punishment in daily life includes a much broader repertoire of punishment behaviors and a more extended variety of social situations in which people enact this repertoire.

Further, and related to the above argument, studies using economic games to examine cooperation typically operationalize punishment as economic sanctioning—i.e., paying a cost to reduce another individual's monetary outcomes. There are two related caveats to this operationalization of punishment. First, it is unclear what sort of real-world behaviors—and, importantly, psychology—such economic sanctioning maps onto. For example, can costly punishment in the lab be considered equivalent to punishment via physical confrontation in the field? Second, if interpreted strictly, lab punishment via economic sanctioning might only correspond to a limited subset of the strategies that people use against offenders in ecologically valid settings.

To capture the breadth of people's responses to norm violations in daily life settings, we define and operationalize punishment broadly. Our definition encompasses both costly strategies of punishment, which involve physical and verbal confrontation directed at the offender, *and* lower-cost strategies of punishment, which involve gossip, exclusion, and avoidance. These various strategies can each inflict costs on offenders, be it via physical threats, verbal condemnation, reputational damage, or the withdrawal of benefits. Many others in the cooperation literature have indeed considered these various means of imposing costs on others as punishment (e.g., punishment as *gossip*, *ridicule*, and *ostracism*, see Boehm, 1993, 2009, Kerr et al., 2009; Raihani & Bshary, 2019; punishment as *communication* and *verbal condemnation*, see Cushman, Sarin, & Ho, 2019; punishment as *benefit withdrawal*, see Balafoutas et al., 2014).

That said, we wholeheartedly agree with the reviewer that our findings should be read and interpreted while keeping in mind our inclusive definition of punishment (rather than the narrower definition of punishment as costly sanctioning in interactions with strangers). We have now made multiple changes to the manuscript to emphasize that (a) laboratory experiments have typically focused on a specific subset of situations and on a very specific strategy of costly punishment (pp. 3-4); (b) we aim to instead capture a broader range of high- *and* low-cost responses to norm violations occurring in interactions within a variety of relationships (p. 4); and (c) in so doing, we consider a whole host of strategies potentially imposing costs on offenders, through physical deterrence, verbal communication, reputation manipulation, and the withdrawal of social benefits (pp. 3-5). In the discussion (p. 14), we (d) acknowledge that by using this broad definition of punishment, we are erring on the side of including behaviors that have not effectively inflicted costs on offenders. Nevertheless, we posit that this limitation is countered by the benefits of the rich information we obtain regarding a variety of different punishment responses in daily settings.

More concerning to me, though, is the direct confrontation question for whether someone took action. Merely confronting an offender about their behavior, without any other qualifiers, doesn't constitute punishment. Looking at some of the examples of norm violations reported (more on this below) causes me a bit of concern. Several of them mention someone being late or dropping out of contact—I really don't see how merely confronting a person about that (e.g., telling them you don't appreciate wasting your time,

*they'd better not do that again in the future, etc.) constitutes punishment. What cost is being imposed? Certainly a confrontation *could* include punishment (verbally attacking the person, withdrawing benefits in retaliation, etc.), but without further clarification I'm pretty unconvinced that just "confrontation" maps onto the construct of punishment focused on in the cooperation literature and what, I presume, the authors here are really interested in.*

One way around this problem could be to incorporate the free response questions that participants answered about what action they took, which would clarify when punishment actually occurred. Without doing something along those lines, I think it's a bit misleading to characterize all forms of confrontation as punishment—if not, perhaps the authors could instead refer to direct confrontation and mention that punishment is a subset of those actions. Going this latter route, the paper could be reframed a bit more toward, say, strategies to recalibrate social norm violations, rather than solely focusing on "punishment."

We thank the reviewer for this comment which helped us realize we could better communicate our operationalization of direct confrontation. When instructing our participants prior to the daily assessment phase, we provided detailed information about the behaviors considered as direct confrontation (see protocol document, available on our OSF pre-registration page; <https://tinyurl.com/y32cunul>). Specifically, we provided concrete examples of what we meant by confronting the offender, which were "*physically stopping this person*" and "*arguing with him/her*". However, we indeed did not restrict participants to *only* report physically or verbally aggressive acts (partly because we see verbal condemnation, even if non-aggressive, as imposing costs on offenders; see our earlier response). In the revised manuscript, we now describe the instructions we gave to participants (pp. 20-21) and clarify that direct confrontation can include verbal condemnation of the sort mentioned by the reviewer (i.e., telling someone that you don't approve of their behavior and that you would like them to change it in the future; p. 3).

More importantly, we take the reviewer's point to suggest caution, especially in interpreting descriptive findings regarding the *prevalence* of direct confrontation in daily life—a point of contention in the cooperation literature. We agree that the rates of direct confrontation we observed should be interpreted while keeping in mind how our study deviates from narrower operationalizations of direct punishment. Indeed, these deviations could well be responsible for the fact that we observe higher rates of confrontational punishment than other studies, which have tried to stick closer to the operationalizations of punishment in second- and third-party punishment games (e.g., Balafoutas et al., 2014; Pedersen et al., 2019).

To address this issue, we went back to our data and checked the prevalence of direct confrontation in situations that more closely map onto (1) second-party punishment games (i.e., self-relevant offenses in which the perpetrators are strangers) and (2) third-party punishment games (other-relevant offenses in which perpetrators and victims are strangers). When we only look at (1) self-relevant offenses committed by strangers ($k = 403$), which are arguably closest to the lab situation

of a second-party punishment game, gossip (in 48.9% of events) and social avoidance (in 47.4% of events) are almost equally prevalent, whereas direct confrontation occurs less often (in 34.2% of events). Note, however, that the rate at which direct confrontation occurs is similar to the rate of confrontation in the overall sample.

Importantly, when we look at (2) other-relevant offenses that are committed by strangers *and* victimize strangers ($k = 136$), which are arguably closest to the lab situation of a third-party punishment game, we observe substantially lower rates of direct confrontation (in 11.8% of events) compared to gossip and avoidance (in 39.0% and 36.0% of events, respectively). That said, only a small number of daily offenses fit this criterion. Nevertheless, when we only consider these situations, the prevalence of direct confrontation is much lower than in the overall sample, consistent with other work suggesting that direct third-party punishment is rare in field settings (see Guala, 2002; Pedersen et al., 2019). We now mention these findings in the manuscript (p. 7) and report them in detail in the SI (pp. 4-5).

At this stage, we have not analyzed the free response descriptions of punishment behaviors. This is largely due to a methodological caveat: in each daily survey, we only provided one text field for participants to describe *any* behavior they engaged in. Thus, participants could have responded that they engaged in multiple punishment behaviors (which were not mutually exclusive) but could have then described only one or some of them in the open-ended responses. We think that this issue limits the conclusions we can draw from our qualitative data. That said, if the editor and reviewers prefer a different approach than the one we took here, we are happy to consider it. For illustrative purposes, we now include examples of free response descriptions of each type of punishment behavior (randomly chosen after selecting cases in which participants indicated *only* engaging in direct confrontation, gossip, or social avoidance; see SI, pp. 28-32).

*- From my reading of the methods and the supplemental material, it seems like the authors did not exclude any cases from their analyses on the basis of the type of norm violation reported. If this is true, I am a bit concerned about the relevancy of some of the examples of social norm violations that are reported in the supplemental material. Some examples that caught my eye: "I'm in a hotel and filled up my 2 bottles of water during the breakfast buffet. I heard a man at the table next to me say to his table companions that this wasn't allowed", "I was working when a man without shirt came to ask things. Some other quests were looking uncomfortable", "When I was parking this afternoon, some other driver deemed it necessary to honk loudly. He probably thought I wasn't fast enough", "He tried to flirt with me and I didn't take it kindly", and "My partner wanted to be of too much help." Perhaps there are some cultural differences at play here (I'm American), but I find examples like these to be a decent stretch from even what I would colloquially call a social norm violation, much less a *cooperative* social norm violation that much of the cooperation literature is focused on, which is how the authors set the context of the paper in the introduction.*

The reviewer is correct; we have not excluded any cases based on the type of norm violation reported. As with the elicitation of punishment responses, when instructing participants about the types of events we were interested in, we provided detailed, theory-based definitions of what we consider norm violations. In short, we asked participants to think about others' behaviors that were immoral, unacceptable, or improper; that they considered as going against their values or principles; and that could potentially lead to disapproval and punishment from themselves and others (for more details, see p. 19, lines 400-405 and protocol document on the OSF, <https://tinyurl.com/y32cunul>). Thus, our method to elicit norm violations was designed to include situations that are typically studied in the cooperation literature (social dilemmas, i.e., situations that involve a conflict between individual and collective interests), without excluding other interdependent situations in which people can violate norms. Specifically, our definition includes violations of convention norms such as those mentioned by the reviewer (e.g., making rude or inappropriate remarks, displaying nudity in public, responding angrily while driving). Again, we believe that this comprehensive approach represents a strength of our method in that it allows us to assess punishment responses to a variety of ecologically valid situations, rather than focusing only on violations of cooperative norms in social dilemma situations. That said, we understand that the emphasis we put in reviewing the cooperation literature might mislead some readers into thinking we *only* look at situations that are equivalent to social dilemmas. In order to avoid this misunderstanding, we now provide our definition of norm violations in the introduction (p. 4).

- *Why did the authors distinguish between physical and verbal motivations for punishment but lump these together for behavior?*

This decision was due to practical constraints. We tried to minimize the amount of time participants had to spend on each daily survey, and thus limit participant fatigue and dropout. This is an important consideration when designing studies in daily life (Conner & Mehl, 2011), where participants are asked to fill out multiple daily assessments over the course of weeks. Based on the intuition that physically aggressive behaviors are quite rare in daily life—and especially in the society we sampled—we decided not to include an item for physical attack, thus reducing time spent by participants on each daily survey.

- *The authors raise a good point that their daily sampling method can reduce recall bias relative to recall methods. I totally agree. However, the prospective design that the authors use here does have a weakness that they don't address: by instructing participants at the outset that they will need to report on norm violations and their responses to them every day, I would imagine there is some effect almost like an intervention. Particularly given that taking action in response to norm violations is socially desirable, I would suspect that the design employed here both (a) caused participants to be more vigilant for norm violations in their everyday life than they would normally, and (b) be more likely to take action than they normally would be since they would know they would have to report a lack of action at the end of the day if they didn't. Given that the main focus of the paper is on the use of different punishment strategies under different circumstances (i.e., the*

meat of the paper is looking at differences among contexts, not absolute levels of punishment), I don't think this is a problem that undermines the main findings. I do think it's worth mentioning, however, that it's a possible limitation—and I do also think it has likely resulted in higher levels of action taken than one would observe more naturally (i.e., without participants being conscious of their intervention behavior being monitored).

We agree with the reviewer that the prospective design we used has these limitations, and we now mention this methodological limitation in our discussion (p. 14).

- Very minor, but some of the language used to describe motivations is hard to parse. One particular example on p.6, “We found that the strength of motivations varied across types of punishment” makes it seem like they are referring to motivations either in response to types of punishment or across different scenarios. Really, they are talking about motivations to engage in different types of punishment, so I'd consider rephrasing (I kept running into this in the results sections in both the main text and supplemental). Again, not a big issue but I think it could make the manuscript clearer to touch it up a bit.

Following this and an earlier comment by the reviewer, we have changed the phrasing in all sections reporting results on motivations to punish offenders. We hope that the current presentation of findings is clearer.

To reiterate, I think this is a very important contribution to the literature and should be quite high impact if the authors are given the opportunity for a revision.

We thank the reviewer once more for the kind remarks and the constructive feedback.

Reviewer #2

My recommendation to publish is unconditional: the authors do not need to change anything. Below, I have a few thoughts (and minor typos).

We are very grateful to the reviewer for the positive and supportive remarks.

Main paper:

Line 146 When you refer to the “second party”, I assume you mean the victim of the offense (with the offender as the first party). If so, you might want to say that in parentheses.

We appreciate the suggestion. We have changed that sentence to read “multiple recent vignette studies find that *victims of norm violations (i.e., second-parties)* are more motivated to directly punish offenders than are third-party observers.” (change in italics)

Line 192 This is just my curiosity. On the weaker association between power and motivation to confront, I wonder what would happen controlling for closeness/ how

valuable the offender is to the subject. Is the weaker association because you are more motivated to confront people you are close too? (another result of yours)

We thank the reviewer for this suggestion. We ran new analyses to explore this possibility. First, we re-ran a model testing the effects of power on motivations to punish offenders, this time controlling for participants' WTR towards offenders. When controlling for WTR, we observed no effects of situational power on motivations to punish offenders (power \times punishment type interaction, $p = .327$; main effect of power, $p = .107$). Second, we re-ran our original model testing the effects of power on motivations to punish offenders, this time controlling for emotional closeness towards offenders. When controlling for closeness, we observed the same pattern of null effects as in the model with WTR. Thus, when accounting for the valuation of offenders or emotional closeness, we observed that people were similarly motivated to punish offenders irrespective of their situational power.

Next, we also wondered if people were similarly likely to follow through with punishment (i.e., engage in punishment behaviors), irrespective of their relative power. This was not the case. We re-ran a model testing the effects of power on punishment behaviors, this time controlling for WTR towards offenders. Results from this model replicate the pattern of results described in our manuscript. After controlling for WTR, participants' power was differentially associated with distinct punishment behaviors (Wald $\chi^2(2) = 25.70, p < .001$). When participants had more power, they were more likely to engage in confrontation ($b = 0.41$, Wald $\chi^2(2) = 24.74, p < .001$), compared to gossip ($b = -0.54$, Wald $\chi^2(2) = 23.17, p < .001$) and avoidance ($b = -0.50$, Wald $\chi^2(2) = 18.34, p < .001$). Finally, we tested the effects of power on punishment behaviors, this time controlling for emotional closeness towards offenders, and observed a similar pattern of results as in our main analyses and above.

We describe these findings in more detail in the SI (p. 15-18).

Lines 474, 476 run (not ran)

We appreciate the reviewer catching the error – we have changed this.

SI 202 “they endorsed less indirect compared to confrontational motivations” The use of less here makes a little garden path. Maybe: they endorsed indirect motivations less than confrontational ones.

We have changed the phrasing to “they endorsed motivations to punish via indirect means less so than motivations to punish via confrontational means.” We thank the reviewer once more for the careful read and positive feedback.

References

- Archer, J., & Coyne, S. M. (2005). An integrated review of indirect, relational, and social aggression. *Personality and Social Psychology Review*, 9, 212-230.
- Balafoutas, L., Nikiforakis, N., & Rockenbach, B. (2014). Direct and indirect punishment among strangers in the field. *Proceedings of the National Academy of Sciences*, 111, 1592415927.
- Boehm, C. (1993). Egalitarian behavior and reverse dominance hierarchy. *Current Anthropology* 34, 227-254.
- Boehm, C. (2009). *Hierarchy in the forest: The evolution of egalitarian behavior*. Harvard University Press.
- Campbell, A. (1999). Staying alive: Evolution, culture, and women's intrasexual aggression. *Behavioral and Brain Sciences*, 22, 203-214.
- Conner, T. S., & Mehl, M. R. (2011). *Handbook of Research Methods for Studying Daily Life*. Guilford Press.
- Cushman, F., Sarin, A., & Ho, M. K. (2019). Punishment as communication. Pre-print available at PsyArXiv: <https://psyarxiv.com/wf3tz>
- Eisenberger, N. I., Lieberman, M. D., & Williams, K. D. (2003). Does rejection hurt? An fMRI study of social exclusion. *Science*, 302, 290-292.
- Guala, F. (2012). Reciprocity: Weak or strong? What punishment experiments do (and do not) demonstrate. *Behavioral and Brain Sciences*, 35, 1-15.
- Kerr, N. L., Rumble, A. C., Park E., Ouwkerk, J. W., Parks, C. D., Gallucci, M., & Van Lange, P. A. M. (2009). How many bad apples does it take to spoil the whole barrel?: Social exclusion and toleration for bad apples. *Journal of Experimental Social Psychology*, 45, 603-613.
- Pedersen, E. J., McAuliffe, W. H., Shah, Y., Tanaka, H., Ohtsubo, Y., & McCullough, M. E. (2019). When and why do third parties punish outside of the lab? A cross-cultural recall study. *Social Psychological and Personality Science*. Advance online publication: 1948550619884565.
- Raihani, N. J., & Bshary, R. (2019). Punishment: one tool, many uses. *Evolutionary Human Sciences*, 1, e12.
- Williams, K. D. (2007). Ostracism. *Annual Review of Psychology*, 58, 425-452.
- Yamagishi, T., Jin, N., & Kiyonari, T. (1999). Bounded generalized reciprocity: Ingroup boasting and ingroup favoritism. *Advances in Group Processes*, 16, 161-197.

***REVIEWERS' COMMENTS:

Reviewer #1 (Remarks to the Author):

I was Reviewer 1 on the initial round of reviews. As I mentioned last time, this is a fantastic paper that I think will have a large impact. Most of my concerns last time were just semantic and I think the authors have done a fantastic job addressing all of them. I enthusiastically endorse publishing the current manuscript as is.

Reviewer #2 (Remarks to the Author):

This paper is, if anything, even better than before. It should be published. The authors' response to reviewers comments was compelling; they have responded to my satisfaction to both reviews. My assessment is as before:

"This is an excellent paper, with a unique dataset and important results on a hotly debated issue in the many communities that study the evolution of cooperation and its psychological foundations (especially: evolutionary biologists, anthropologists, and psychologists; behavioral economists; cultural evolution modelers). Whether people punish others, when, and how is a central issue in these debates, and the answer matters to theories of whether cooperation evolved via reciprocity (and by partner choice versus partner control, i.e., punishment), via "strong reciprocity"/ group selection, via cultural group selection, or via some other population level process. This paper provides data that richly speaks to these theories. It also contributes to theories of the function of different emotions. On top of all this, it is beautiful and clearly written. It should be published with high priority."